# Effect of Multiple Reclosing Time Intervals on Axial Vibration of Winding

Lu Sun [1,*], Shuguo Gao [1], Yuan Tian [1], Ruidong He [2], Fuyun Teng [3], Liang Wang [3], Jianghai Geng [3], Ping Wang [3], Xinyu Wang [3], Zikang Zhang [3], Jianhao Zhu [3], Jiaxin Yao [3] and Yufei Yao [3]

[1] Hebei Electric Power Research Institute, Shijiazhuang 050021, China; dyy_gaosg@he.sgcc.com.cn (S.G.); dyy_tiany1@he.sgcc.com.cn (Y.T.)
[2] State Grid Hebei Electric Power Co., Ltd., Shijiazhuang 050057, China; nanwangly@126.com
[3] Hebei Provincial Key Laboratory of Power Transmission Equipment Security Defense, North China Electric Power University, Baoding 071003, China; 220222213028@ncepu.edu.cn (F.T.); walar2017@163.com (L.W.); gengjianghai@163.com (J.G.); pingwang0501@163.com (P.W.); wangxy1996@ncepu.edu.cn (X.W.); xtyz201110@163.com (Z.Z.); 18953423503@163.com (J.Z.); zz261923@163.com (J.Y.); 18135910183@163.com (Y.Y.)
* Correspondence: hbdky_sunlu@163.com; Tel.: +86-136-0321-9753

**Featured Application: The research provides a theoretical reference for transformer closing-control strategies, in which the superposition effect decreases as the reclosing interval increases.**

**Abstract:** When a transformer suffers a permanent fault, it will suffer a short-circuit impulse again after reclosing. If the previous vibration of the winding is not attenuated completely and the winding is subjected to a secondary impulse within a short time, the secondary vibration response will have a superposition. The aim of this study was to analyze the effect of the anti-short-circuit ability of operational transformers subjected to a secondary short-circuit current impulse. In this paper, a model is established for calculating axial vibration in transformer windings and effects on the vibration response of windings under different closing phase angles and short-circuit intervals are analyzed. The results show that the vibration acceleration of windings is a V-shaped variation at phase angles from 0° to 180°, reaching the maximum values at 0° and 180° and reaching the minimum value at 90°. When the transformer recloses on a permanent short circuit, due to the superposition effect, the vibration acceleration amplitude of the secondary impulse will be greater than that of the primary impulse, but as the reclosing interval increases, the superposition effect decreases continuously. When the interval is 600 ms, the superposition effect for the vibration acceleration of the secondary impulse attenuates to 83.3%. The superposition effect is not significant after 600 ms. The research provides a theoretical reference for transformer closing-control strategies.

**Keywords:** multiple reclosing; winding vibration; effect of superposition

## 1. Introduction

In the operation of a power system, the transformer will automatically reclose after an initial short circuit [1]. If the fault is transient, the system will resume normal operation [2]. However, following permanent faults, the transformer will shortly undergo a second impulse of a short-circuit current [3]. In the reclosure-failure case the winding experiences a cumulative superposition effect of multiple impulses over a short time period, when the previous impulse is not eliminated and the subsequent impulse occurs [4]. This effect leads to greater deterioration of the winding [5]. When the short circuit is removed, the transformer structure fails to fully recover, and the insulating pad between copper disks is deformed by vibration [6]. Additionally, the short-circuit current and magnetic flux distribution change [7]. With increased understanding of the cause and process of short-circuit damage in transformer winding [8], transformer winding vibration in multiple short-circuit impulses has attracted increased attention and research by scholars [9].

H. L. proposed a laminated reinforcement cylindrical-shell model to study the deformation impulses of multiple short circuits [10]. Wang Huan found the elastoplastic tangent stiffness matrix of transformer winding elements according to the large displacement elastoplastic theory [11]. Zhang Chi established a refined model for the middle disk of the transformer's medium-voltage winding [12]. Du Jian proposed a method for transformer model creation suitable for direct coupling calculation of the electromagnetic field and structure field [13]. Then, a numerical calculation model incorporating multiple short circuits of transformer winding was used to analyze the effect of different residual stresses on the residual deformation [10]. The distribution and variation trends in winding-shape variables, equivalent stress, and equivalent plastic strain of loading and unloading short-circuit force in a single short-circuit impulse were obtained using Workbench. The accumulation mechanism of winding axial vibration in multiple short-circuit impulses was obtained [14]. When the transformer is subjected to the accumulation of short-circuit impulses, the winding undergoes cumulative deformation [12].

These studies established the transformer-winding vibration model and calculated the vibration response of the winding. Based on this work, the cumulative effect of winding vibration was analyzed and the vibration mechanism of the winding multiple short circuit was proposed. However, the established model does not consider the influence of disk displacement and acceleration of the unattenuated short circuit on the winding structure and other parameters, nor does it account for the superimposed influence of other parameters on the vibration response of the secondary short circuit.

In this paper, the mathematical model of transformer axial vibration is established based on an actual 110 kV transformer. Furthermore, the characteristics of variation in vibration amplitude, vibration displacement, and acceleration are obtained. Considering the continuous influence of winding vibration, the reclosing time interval is changed, the winding's vibration response at the corresponding time interval is calculated, the behavior of the winding's secondary vibration at different time intervals is analyzed, and the shortest reclosing time is judged.

## 2. Dynamic Model of Winding Axial Vibration

Generally, transformer vibration includes core vibration and winding vibration. Core vibration occurs by magnetostriction and relates to voltage [15]. Winding vibration is generated by winding current and leakage magnetic field [16]. When transformers are short-circuited, the winding current may reach dozens of times the rating of normal operating current and the primary side voltage drops rapidly. At that point, the winding vibration will be much stronger than the core vibration [17] and the accumulation of axial vibration of winding becomes more significant.

### 2.1. Solving the Axial Vibration Model

The transformer winding axial vibration system is a linear vibration system. A linear vibration system consists of mass, damping, and spring stiffness. In the system, the disks are regarded as a lumped mass, the insulating pad between the disks and the pressure plate is regarded as an elastic element, and the oil-filled area between the disks is regarded as damping. Therefore, the transformer winding can be abstracted as a "mass-damping-spring" model and the solution method of the response of the transformer winding axial vibration is an analytical method. The model is presented in Figure 1 [18].

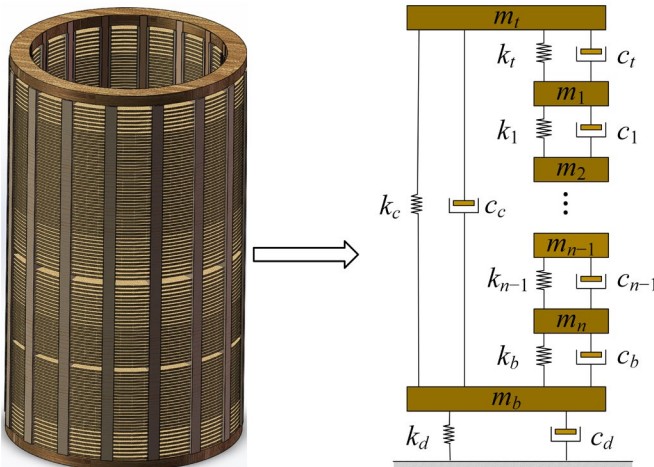

**Figure 1.** Axial mass-damping-spring model of winding.

In Figure 1, the mass of the individual disk is $m_i$, $i$ = 1, 2, ..., $n$; $n$ is the number of winding turns; the equivalent stiffness of the insulating pad between $i - 1$-th disk and $i$-th disk is $k_i$; and the equivalent viscous damping of the insulating oil between the lines is $c_i$ [19]. According to the "mass-damping-spring" model of the actual transformer and force analysis, the vibration of each disk can be expressed by a second-order differential equation system, which is as follows:

$$
\begin{cases}
m_t \ddot{x}_t = k_t(x_1 - x_t) + c_t(\dot{x}_1 - \dot{x}_t) + k_t(x_b - x_t) + c_c(\dot{x}_b - \dot{x}_t) - F_c - m_t g \\
m_1 \ddot{x}_1 = k_t(x_t - x_1) + c_t(\dot{x}_t - \dot{x}_1) + k_1(x_2 - x_1) + c_1(\dot{x}_2 - \dot{x}_1) - F_c - m_1 g + f_1 \\
m_2 \ddot{x}_2 = k_1(x_1 - x_2) + c_1(\dot{x}_1 - \dot{x}_2) + k_2(x_3 - x_2) + c_2(\dot{x}_3 - \dot{x}_2) - F_c - m_2 g + f_2 \\
\quad \cdots \cdots \\
m_n \ddot{x}_n = k_{n-1}(x_{n-1} - x_n) + c_{n-1}(\dot{x}_{n-1} - \dot{x}_n) + k_b(x_b - x_n) + c_b(\dot{x}_b - \dot{x}_n) - F_c - m_n g + f_n \\
m_b \ddot{x}_b = k_b(x_n - x_b) + c_b(\dot{x}_n - \dot{x}_b) + k_c(x_t - x_b) + c_c(\dot{x}_t - \dot{x}_b) - k_d x_b - c_d \dot{x}_b - F_c - m_b g
\end{cases}
\tag{1}
$$

where $m_n$ is the mass of the $n$-th disk; $n$ is the order of the disk from top to bottom; $m_t$ is the mass of top platen, $m_b$ is the mass of the bottom platen; $k_t$ and $c_t$ are the equivalent stiffness of the insulating pad and damping of oil between the top platen and the first disk; $k_b$ and $c_b$ are the equivalent stiffness of the insulating pad and damping of oil between the bottom platen and the last disk; $k_c$ is the equivalent stiffness of the winding brace and external compression structure; $c_c$ is the corresponding equivalent damping coefficient; $k_d$ and $c_d$ are the equivalent stiffness of the insulating pad and damping of oil between the bottom platen and the transformer base; $x_i$ is the vibration displacement of the $i$-th disk; and $f_i(t)$ is the electric force applied to the $i$-th disk.

Compared with the finite element simulation model, the calculation speed of this model is faster under the same mesh. In contrast to the conventional mass-damping-spring model, the equivalent stiffness and damping of the fixed parts of the winding braces and top and bottom platens, as well as the equivalent stiffness and damping of the bottom telos of the winding and the transformer base, are considered in this model. Additionally, the influence of each part on the axial vibration of the winding is considered more comprehensively.

To facilitate the subsequent calculation, Equation (1) is written in matrix form:

$$
M\{\ddot{x}(t)\} + C\{\dot{x}(t)\} + K\{x(t)\} = \{f(t)\}
\tag{2}
$$

where $M$ is the $n$-order matrix of disk mass, $C$ is the $n$-order matrix of damping, $K$ is the $n$-order matrix of stiffness, $\{\ddot{x}(t)\}$ is the n-dimensional vibration acceleration column vector, $\{\dot{x}(t)\}$ is the vibration $n$-dimensional velocity column vector, $\{x(t)\}$ is the $n$-dimensional displacement column vector, and $\{f(t)\}$ is the $n$-dimensional electric force column vector.

In the case of a transformer short circuit, to facilitate the analysis, only periodic and transient components of the short-circuit current are considered [20]:

$$I = I_m\left(e^{-\frac{R}{L}t} - \cos\omega t\right)$$ (3)

where $I$ is the steady short circuit current; $I_m$ is the amplitude of the alternating component of short-circuit current; $L$ is the short circuit equivalent inductance of the system; and $R$ is short-circuit equivalent resistance of the system [21]. Then, the electromagnetic force $F$ acting on the winding is

$$F = \frac{1}{2}pI_m^2\left(e^{\frac{-2R}{L}t} - 2e^{\frac{-R}{L}t}\cos\omega t + \frac{1}{2}\cos 2\omega t + \frac{1}{2}\right)$$ (4)

where $p$ is the electromagnetic force coefficient and the electromagnetic force frequency is twice $\omega$, which is the current frequency [22].

The system characteristic formula is

$$\left|K - \omega_n{}^2 M\right| = 0$$ (5)

where $\omega_n$ is the natural frequency of the $n$-order mode shape.

The natural frequencies of the $r$ order are substituted into the eigenvalue problem formulas of the system successively.

$$K\{u\} = \omega_n{}^2 M\{u\}$$ (6)

where $\{u\}$ is the modal row vector.

### 2.2. The Parameters of Transformer Model

Generally, a pie structure is used in a 110 kV three-phase transformer. The pie structure means that the copper-wire turns are continuously wound into a disk and several disks are connected, stacked, and separated by pads to form a phase winding in the axial direction. Parameters of the vibration model are set according to the actual transformer parameters.

In the axial direction of windings, the top and bottom platens are fixed at the telos of the transformer winding and the insulating pads are stacked between adjacent disks or platens. The support stiffness depends mainly on the insulating pad. The stiffness of the insulating pad can be expressed in the following formula:

$$k = \frac{abE}{h}$$ (7)

where $E$ is the elastic modulus of the insulating pad, which is $7.69 \times 10^3$ Mpa; $a$ is the width of an insulating pad; $b$ is the distance from the inside to the outside of the disk surface; and $h$ is the height of the insulating pad, which is equivalent to the distance between adjacent disks or platens.

For a more detailed analysis of the whole winding's vibration, we use individual disks, rather than the entire coil. An individual disk mass can be calculated as follows:

$$m = \frac{LnS\rho}{1000N}$$ (8)

where $m$ is the mass of individual disk; $L$ is the total wire length of a single phase; $n$ is the number of turns of a single disk; $S$ is the sectional area of a single wire; $\rho$ is the density of disk material; and $N$ is the number of winding turns [23]. It is thus possible to determine the quality of every winding disk.

The damping of the equation system is calculated using three-thousandths of the stiffness matrix. According to the parameters of the actual transformer structure and Equations (7) and (8), the calculation results of model parameter are shown in Table 1.

**Table 1.** Vibration model parameters.

| Parameters | Value |
| --- | --- |
| $m_t$, $m_b$/(kg) | 10.8 |
| $m_1-m_{107}$/(kg) | 53.8 |
| $k_7-k_{26}$, $k_{80}-k_{99}$/(N·m$^{-1}$) | $3.86 \times 10^9$ |
| $k_{27}$, $k_{79}$, $k_{107}$, $k_t$, $k_d$/(N·m$^{-1}$) | $1.61 \times 10^9$ |
| $k_{29}-k_{40}$, $k_{66}-k_{78}$/(N·m$^{-1}$) | $3.22 \times 10^9$ |
| $k_{41}-k_{49}$, $k_{57}-k_{65}$/(N·m$^{-1}$) | $2.41 \times 10^9$ |
| $k_{28}$/(N·m$^{-1}$) | $2.15 \times 10^9$ |
| $k_{53}$/(N·m$^{-1}$) | $1.48 \times 10^9$ |
| $k_c$/(N·m$^{-1}$) | $1.38 \times 10^{11}$ |

## 3. Effect of Reclosing Interval on Winding Vibration

After an external short circuit of the transformer, the circuit breaker will cut off the fault current. After a certain time interval, the reclosing device will attempt to reclose [24]. This time interval is the reclosing time interval. When a short-circuit fault occurs outside the transformer, a large electromagnetic force will be generated in the winding. This force will cause the reciprocating vibration of the winding in the axial direction, cause the deformation of the winding, and change or even collapse the winding structure [25]. The changes undergone by each disk in the winding when the external short circuit occurs can be calculated according to the model in the above section.

### 3.1. Characteristics of Winding Axial Vibration

According to the numerical relationship between the stiffness of the system and the damping matrix, the winding stiffness is several thousand times the equivalent damping. Then, the system of winding axial vibration is a rigid system, and the ode23t solver on 23.2.0.2365128 (R2023b) is used to find the value.

By substituting the values in Table 1 into the matrices **M** and **K**, the four mode frequencies of the axial vibration of the winding and the corresponding mode shapes are calculated. These values are shown in Figure 2. The first mode shape of the winding is the axial vibration of each coil. The second mode frequency is double frequency, and the corresponding mode shape is the reverse vibration of the upper and lower parts of the winding. The 3rd and 4th mode shapes of the windings show the same or reverse vibration of the disks at different positions along the axis of the windings [26].

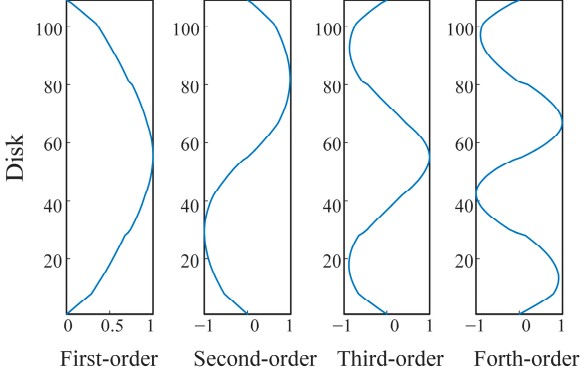

**Figure 2.** Modes of each order of winding.

At the beginning of the winding short-circuit vibration, the transient vibration component and the steady vibration component coexist and the overall vibration of the winding is the result of the superposition of the two vibration components. The axial vibration of each disk in the transient state is mainly the damped forced vibration in the excitation of electromagnetic force. Figure 3 shows the motion curve of the axial damped forced-vibration system of the disk of the winding.

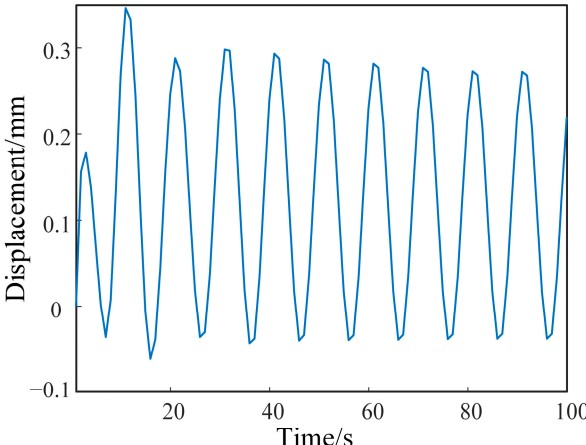

**Figure 3.** The displacement change of the upper-layer disk.

At 10 ms, the amplitude of vibration displacement is at the maximum. At that moment, it is possible to better observe the displacement change of each disk in the axial direction of windings, so the time selected for observing a vibration shape is 10 ms. The displacement and acceleration changes at 10 ms are shown in Figure 4.

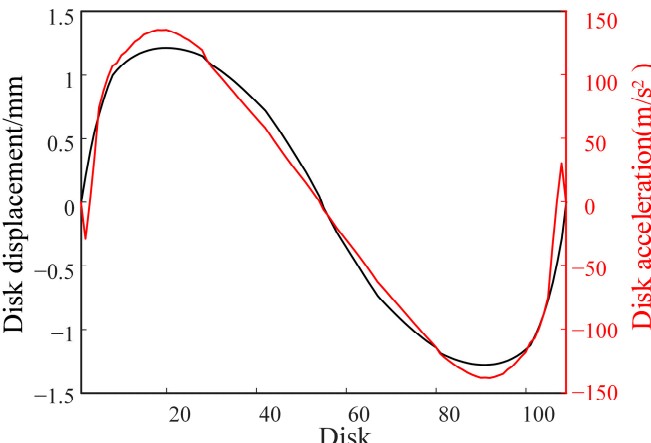

**Figure 4.** Displacement and acceleration of each disk.

The vibration displacement and acceleration waveform in the axial direction of winding is a horizontal "S" shape, and its amplitude reaches a peak at 1/4 and 3/4, but the displacement at the telos and the middle is very small. It roughly corresponds to the law of the second-mode shape, and the second-mode frequency is 100 Hz. This result means that the displacement distribution of the winding in the axial direction follows the theoretical calculation.

When a power transformer is short-circuited, the winding is a system of damped multiple degrees of freedom, and the electrical force is mainly the double-power frequency of 100 Hz. The distribution of vibration displacement and acceleration caused by the electric force applied to the disk is consistent with the second-order modal shape. The amplitude of winding displacement reaches a peak at positions of 1/4 and 3/4, and the vibration displacement at the telos and the middle of winding are very small. The amplitude and motion law of winding vibration are related to the physical properties, current, and frequency of the system itself and have little relationship with the initial state of winding motion [27].

### 3.2. Vibration Response in Different Short Circuit Durations

To study the effect of the short-circuit interval on winding vibration, different short-circuit cut-off times were analyzed. Three disks of different axial heights, equally spaced in the direction of the low-voltage winding axis, were taken as measurement points. The vibration acceleration is shown in Figure 5.

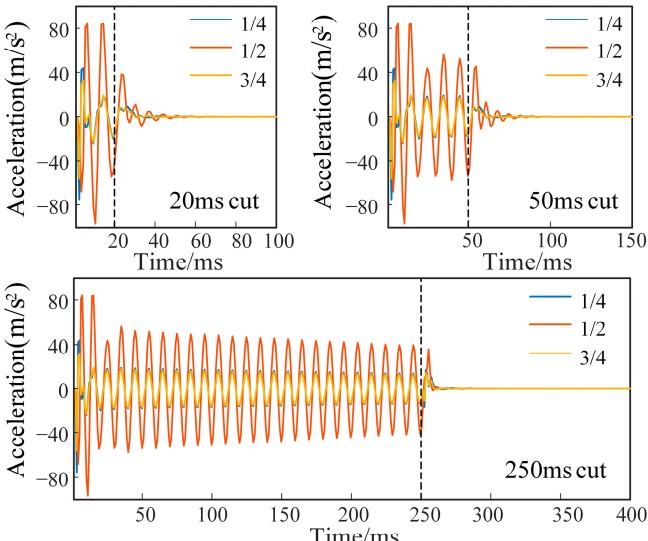

**Figure 5.** Change of Acceleration at different cut-off times.

The amplitude of the vibration acceleration at the middle disk is much larger than that at other positions before the circuit breaker trips. When the circuit breaker is tripped, the action time is short, which causes the acceleration amplitude of the disk to increase sharply. After the breaker trips, the acceleration amplitude of all disks is quickly reduced and vibration acceleration is also attenuated over time. After the circuit breaker trips, the acceleration of the disk at three positions of the winding decays to less than 10% of the acceleration during the circuit-breaker action within 30 ms. When the duration of the circuit-breaker removal is prolonged, the aperiodic components at disks in different positions in the acceleration are attenuated.

According to Figure 6, the displacement at 1/4 and 3/4 of the winding axial height is large before the circuit breaker trips and the displacement in the middle is much smaller than that in other positions. Before the action of the circuit breaker, each disk is subjected to damped forced vibration in electromagnetic excitation. After the breaker action, each disk is subjected to damped free vibration and the vibration displacement at different positions is attenuated more slowly than the acceleration amplitude.

The strength of additive effects varies with the current and the winding structure of the transformer, which is subjected to many short-circuit impacts. When the short-circuit current is small, the superposition of the cardboard is not obvious. With the short-circuit current increasing, the short-circuit electric force on the winding increases and the nonlinear effect of the paperboard is enhanced. The winding loosening caused by the superposition of plastic deformation of the paperboard becomes the main factor influencing the characteristics of the winding vibration.

### 3.3. Vibration Response under Different Short Circuit Intervals

After a short circuit, the windings vibrate strongly, resulting in relaxation of the winding, and the height of the winding is reduced [28]. Even a relaxed winding will not immediately recover after the short circuit is removed [29]. Additionally, the equivalent stiffness of the winding will change when the axial height of the winding falls. It is necessary to wait before the axial height of the winding is restored after a short circuit [30].

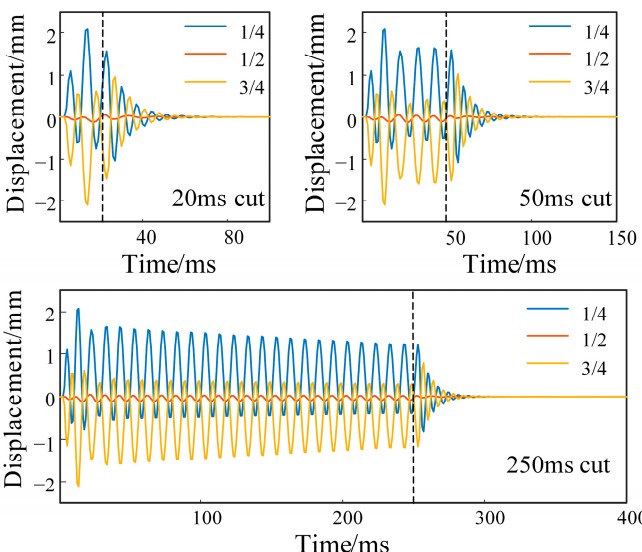

**Figure 6.** Change of displacement at different cut-off times.

At the same time, the breaker trips and the transformer short-circuit current cuts off. At that point, the winding has damped free vibration, showing underdamped attenuation, and the vibration energy is variably attenuated. At different degrees of vibration-energy attenuation, such as at 10 ms and 500 ms after the breaker trips, the breaker closes and a short-circuit current enters the transformer winding. At that time, the winding continues to be subjected to the short-circuit electromagnetic force. Figure 7 shows the disk displacement at different positions.

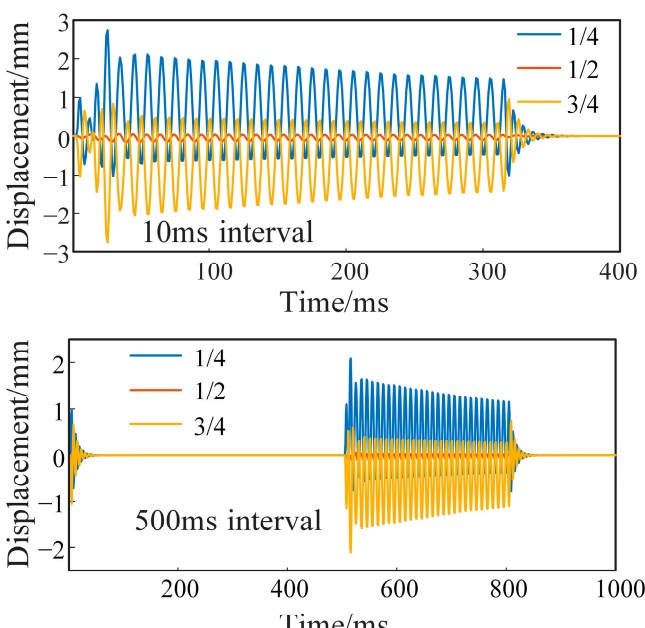

**Figure 7.** Vibration displacement under different degrees of attenuation.

It can be seen from Figure 7 that the vibration-displacement peak value at 500 ms decreases by more than 20% compared with the vibration-displacement peak value at 10 ms. The displacement peak value of the circuit breaker decreases significantly after reclosing. Therefore, as the short-circuit interval becomes larger, the vibration-displacement peak value decreases.

After the breaker trips, the transformer short-circuit current is cut off and the winding's free vibration is damped, showing underdamped attenuation. At different degrees of

vibration-energy attenuation, such as the circuit-breaker action after the trip at 10 ms and 500 ms, the breaker closes and the short-circuit current passes into the transformer winding. At that point, the winding continues to be subjected to the short-circuit electromagnetic force. Figure 8 shows the disk acceleration at different positions.

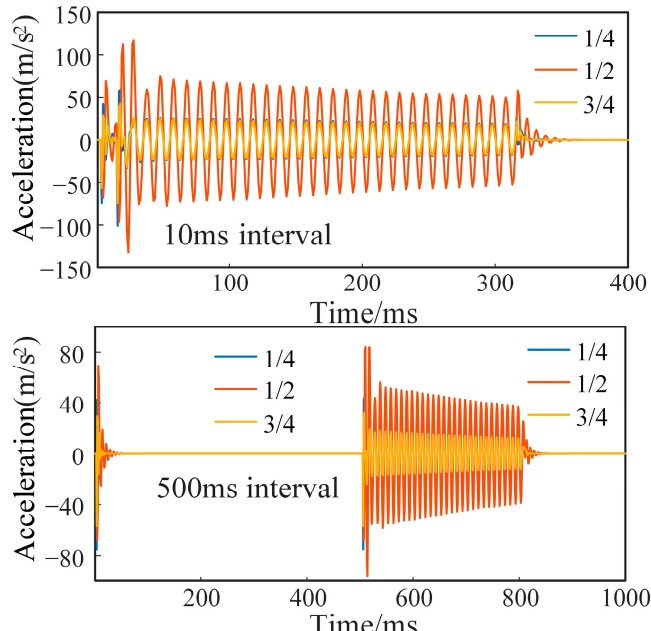

**Figure 8.** Vibration acceleration under different degrees of attenuation.

According to Figure 8, the vibration-acceleration peak value at 500 ms is lower than the vibration acceleration peak value at 10 ms by about 40%. The vibration-acceleration peak value is significantly lower. Therefore, as the short-circuit interval becomes larger, the vibration acceleration decreases.

After 10 ms of vibration attenuation, we selected different phase angles to continue the short-circuit closing, where the closing angle was set to different points in the 0–180° period, each separated by 20°. According to the calculated results of the reclosing short-circuit current, the current through the winding is different at different positions and decays periodically with the change of interval. The electromagnetic force is adjusted by adjusting the secondary short-circuit impulse current, following which the excitation of the winding vibration is changed. The vibration displacement and acceleration of the transformer, in different disks and at different closing phase angles, are shown in Figure 9.

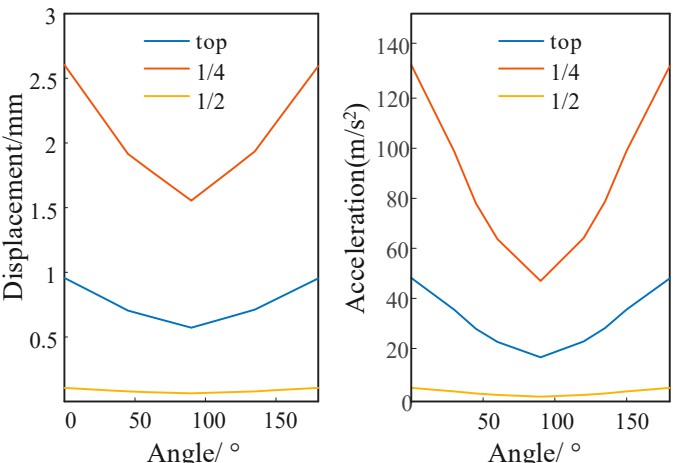

**Figure 9.** Vibration displacement and acceleration under different closing phase angles.

Figure 9 shows the vibration response of the most common single-phase short-circuit fault, which describes the vibration displacement and vibration acceleration law in a closing angle from 0° to 180°. As the phase angle increases from 0° to 90°, the vibration displacement and acceleration response become smaller. However, the vibration displacement and acceleration response become larger as the closing phase angle increases from 90° to 180°. Therefore, the vibration amplitude is the smallest when the closing angle is 90°, and the reclosing phase-angle configuration should be considered.

## 4. Evaluation of the Effect of Reclosing Time Interval on Short-Circuit Vibration

The key factors that contribute to axial vibration in power transformers include the short-circuit electromagnetic force, pretightening force, winding mode, the change of insulating pad structure inside the winding, and the reclosing time interval. When there is a permanent short circuit outside the transformer, the vibration amplitude of the transformer winding also changes with the change in the reclosing time interval. To analyze the corresponding relationship between the two, the following analysis was conducted.

### 4.1. Acceleration Response of Transformer Winding at Different Reclosing Time Intervals

Two parameters describe the intensity of vibration at the transformer secondary short circuit: the vibration acceleration and the vibration displacement. In Figure 7, the variation in vibration displacement in two short circuits is shown. The vibration-displacement peak value of the secondary short circuit close to the permanent short circuit is much greater than that of the vibration-displacement peak value in the case of the primary short circuit, but the vibration displacement requires an additional cycle to reach the peak value. In Figure 8, the variation in vibration acceleration in two short circuits is shown. The acceleration amplitude of the primary short circuit and that of the secondary short circuit are very different, and the change in acceleration is more obvious than the change in displacement. The vibration-acceleration amplitude rises faster when there is damped forced vibration and decreases more significantly when there is damped free attenuation. However, in the same way, the vibration acceleration reaches the peak after a delay of one cycle.

In Figure 9, the interval corresponding to 0–180° is about 10 ms, and the change law of vibration displacement and acceleration in a cycle is nearly identical, with an initial decrease and then an increase within 10 ms. The variation in the range of vibration acceleration at different times is more significant than that of displacement. Therefore, it is easier to obtain clear conclusions by analyzing the acceleration resulting from permanent short-circuit closing. The following analysis considers only the vibration acceleration.

After the transformer short-circuit fault is removed and before closing, the windings are free-vibrating, so closing under different free-vibration intensities will result in different degrees of superposition effect. Figure 10 shows the maximum response of vibration acceleration of the windings during secondary short-circuit impact under different closing waiting times within 0–1000 ms after fault removal.

According to the vibration-acceleration-amplitude curves obtained in Figure 10 in the short-circuit interval from 0 to 1000 ms, the following characteristics of vibration response can be obtained: firstly, due to the superposition effect, the peak of the secondary vibration acceleration is higher than the peak of the primary vibration, but it continues to decay with interval increasing; secondly, the closing waiting time is 0–600 ms and the amplitude of the secondary vibration gradually decays. The amplitude of the secondary vibration becomes essentially stable within 600–1000 ms; the superposition effect is small, and the influence is not obvious.

### 4.2. Calculation of Winding Response Variation at Different Reclosing Intervals

According to the axial dynamic formula of the winding shown in Equation (1), the vibration acceleration of the winding can reflect the magnitude of the resultant force. According to the difference in the vibration acceleration value, a quantitative analysis of the influence of different intervals on the winding is carried out.

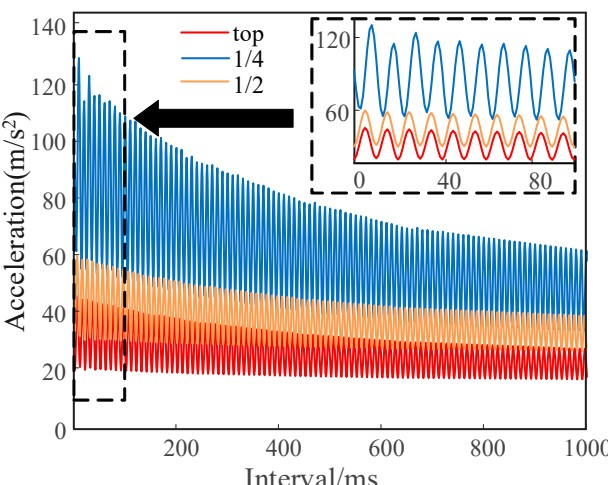

**Figure 10.** Vibration displacement and acceleration at different closing times.

As the short-circuit interval changes, the axial vibration amplitude on the winding changes, as does the attenuation time. The peaks and valleys values in each cycle in the acceleration curve of Figure 10 were sampled and fitted, and the relationship curve between the acceleration amplitude of the secondary vibration and the reclosing interval was obtained, as shown in Figure 11.

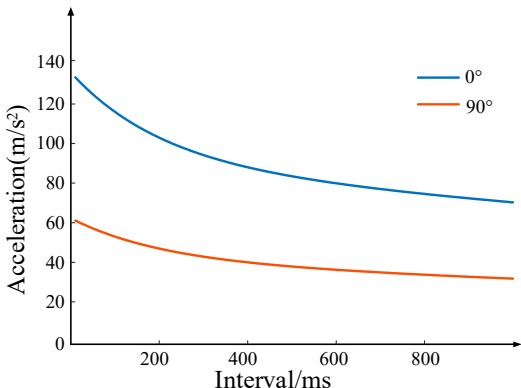

**Figure 11.** Vibration acceleration at different phase angles of 0° and 90°.

The vibration acceleration curve of closing at a 0° phase angle is generally larger than that of closing at a 90° phase angle. Table 2 was obtained by quantitative analysis of the curve in Figure 11. The sampling intervals were 50 ms, 200 ms, 400 ms, and 600 ms.

**Table 2.** Acceleration at different phase angles under different degrees of attenuation.

| Interval | Angle 0° | 90° |
|---|---|---|
| 50 ms | 132.02 m/s$^2$ | 61.13 m/s$^2$ |
| 200 ms | 107.16 m/s$^2$ | 48.97 m/s$^2$ |
| 400 ms | 92.21 m/s$^2$ | 42.19 m/s$^2$ |
| 600 ms | 84.93 m/s$^2$ | 38.37 m/s$^2$ |

As the short-circuit interval increases, the acceleration amplitude of the secondary vibration decreases, and the attenuation reaches a steady state at 600 ms. In a steady state, the vibration acceleration amplitude is reduced by about 40%. The acceleration at 0° is much greater than that at the 90° phase angle. For example, the amplitude at 0° acceleration is more than twice that of the amplitude at 90° in the 50-ms interval. Therefore, adjusting

the closing phase angle can effectively reduce the acceleration amplitude of secondary vibration.

These results can be used to evaluate the quantitative effect of the short-circuit time interval on the transformer and the dynamic relationship between winding vibration and the electromagnetic force. Additionally, the influence of the superposition effect can be more intuitively reflected through this numerical approach. The simulation calculation yields the short-circuit current multiples at which the winding vibration acceleration reaches amplitudes of strong vibration superposition during mechanical recovery to steady state, as shown in Table 3.

**Table 3.** The equivalent amplification factor of short circuit current under superposition effect.

| Interval/ms | 50 | 200 | 400 | 600 | 1000 |
|---|---|---|---|---|---|
| Reclose at 0° | 1.391 | 1.252 | 1.134 | 1.051 | 1 |
| Reclose at 90° | 1.383 | 1.244 | 1.128 | 1.041 | 1 |

In Table 3, the amplification factor of the short-circuit current is can be expressed as the number of times that the initial short-circuit current is magnified on the basis of its original value, when the initial short-circuit vibration acceleration and the second short-circuit vibration acceleration amplitude are equal. Among these values, the short-circuit current amplification multiple is equivalent to the short-circuit current amplification. It can be seen that when the short-circuit time interval reaches 600 ms, short-circuit current amplification is already negligible.

In conclusion, after a short circuit, the acceleration amplitude of the secondary closing decreases as the vibration interval increases. Different reclosing intervals lead to different amplitudes of disk vibration acceleration.

## 5. Conclusions

When a permanent short-circuit fault occurs outside the transformer, the secondary vibration response may be greater than the primary vibration due to the superposition of residual vibrations. In this paper, by establishing an axial vibration model and considering the continuous influence of vibration, the authors were able to study the effect of the reclosing time interval on the vibration response of windings under multiple short-circuit impacts and obtain the following conclusions:

1. When the transformer is subjected to a short-circuit impact, its vibration displacement and acceleration amplitude will change, and the vibration displacement change will be slower than the vibration acceleration. In the case of damped free vibration, the amplitude of the vibration displacement will decay to a relatively low level within 30 ms, and the vibration acceleration will decay to a relatively low level within 25 ms. The vibration acceleration will initially decay at a faster rate, reaching less than 10% of the peak value, but the effect of vibration persists.

2. The response to short-circuit vibration has a period of 10 ms, which corresponds to the range of the current phase angle from 0° to 180°. As the reclosing time interval changes, the amplitude of winding vibration acceleration changes in the phase-angle ranges from 0° to 180°, and the winding vibration acceleration at phase angles of 0° and 180° is much greater than that at a phase angle of 90°. Due to the continuous influence of the last short-circuit vibration, the superposition effect of winding vibration in the second short circuit is more severe than that in the first short circuit. When the reclosing time interval is 10 ms, the peak value of reclosing vibration displacement is 2.74 mm, which is greater than the peak value of the first vibration displacement. At 250 ms, the peak value of secondary vibration displacement is reduced by 20%. In the same case, the peak acceleration of the secondary vibration is reduced by 37%.

3. The equivalent amplification factor of the winding short-circuit current in the secondary closing under the superposition effect is considered by conversion. When the

short-circuit time interval is short, the secondary short-circuit impact is equivalent to the larger short-circuit current effect in the primary short circuit. When the short-circuit time interval is more than 600 ms, the equivalent amplification factor of the closing short-circuit current is less than 1.041 under the ideal closing phase angle of 90° and the superposition effect is negligible.

When the transformer experiences multiple short-circuit reclosings, the closing time interval should not be less than 600 ms to avoid the large vibration impacts, which reduce the life of the transformer. The secondary vibration response of the transformer winding in the axial direction is analyzed in this paper. However, the influence of the continuous vibration of the transformer on the radial component of the secondary vibration of the winding has yet to be established, and the overall vibration-mechanism analysis is still inconclusive.

**Author Contributions:** Conceptualization, L.S. and S.G.; methodology, L.S. and Y.T.; software, R.H.; validation, F.T., L.W. and Z.Z.; formal analysis, J.Z.; investigation, J.Y.; resources, S.G.; data curation, Y.Y.; writing—original draft preparation, L.S.; writing—review and editing, S.G.; visualization, X.W.; supervision, P.W.; project administration, J.G.; funding acquisition, L.S. All authors have read and agreed to the published version of the manuscript.

**Funding:** This research was funded by the Science and Technology Project of State Grid Hebei Electric Power, Fund No. kj2022-022 and the Natural Science Foundation of Hebei Province, Fund No. E2021521004.

**Institutional Review Board Statement:** Not applicable.

**Informed Consent Statement:** Not applicable.

**Data Availability Statement:** Not applicable.

**Conflicts of Interest:** Author Ruidong He was employed by the company State Grid Hebei Electric Power Co., Ltd. The remaining authors declare that the research was conducted in the absence of any commercial or financial relationships that could be construed as a potential conflict of interest.

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
