# Peer review of "Effect of Multiple Reclosing Time Intervals on Axial Vibration of Winding"

_applsci, doi:10.3390/app132111910_

Round 1

Reviewer 1 Report

Comments and Suggestions for Authors

In page 3, first paragraph the symbols have small mistakes, for example ki¬, and it could by k and subindice i. 

ci. c with subindice i, at the same situation in second paragraph with cb. 

In page 4. improve the equations 4 and 6. (parentesis in the functions and close the absolute in the equation respectively)

In page 5. The position of the table 1, it could be better after the citation. 

In page 8, first paragraph, it could be better if the figure have a number. 

Reviewer 2 Report

Comments and Suggestions for Authors

The manuscript is interesting and consistent with scientific art. As part of the research, the authors presented a model of transformer winding vibrations and calculated the winding vibration response to the multiple short-circuit vibration mechanism. The manuscript adopts a transformer axial vibration model based on a real 110kV transformer. The correctness of the proposed vibration model was verified by modal analysis. The authors calculated the responses to the displacement and acceleration of winding vibrations with different durations and reconnection intervals, and analyzed the impact of different short-circuit phase angles and short-circuit superposition effects on the secondary winding vibrations.

The scientific topic discussed in the manuscript seems attractive and necessary. The contribution of the authors' scientific work is large, but requires further actions to make it more attractive and more interesting to the reader. The authors absolutely need to improve:

1. Chapter "Discussion" the authors must clearly analyze the obtained simulation results so that the reader can formulate the impact of the obtained results on the operation of the transformer

2. The "Conclusions" chapter is not included in this manuscript. This seems a bit strange because in the discussion chapter there are the results of the authors' research work, but they are only articulated without a proper discussion, and they should be more developed and discussed in more detail. However, "Conclusions" were completely omitted. The authors must add this chapter, formulate the conclusions obtained from the research work and refer to the results obtained and further research work.

3. Enlarge the literature review because with such a complex and interesting problem discussed, 21 publications are too few. The authors must make a greater revision of research achievements in the topic under consideration, as evidenced by the expansion of the topic in the "Introduction" chapter and the increase in the amount of analyzed literature.

Reviewer 3 Report

Comments and Suggestions for Authors

Overall, the manuscript discusses the effect of multiple reclosing times of transformers . An interesting topic. However, I have some comments on it:

1- Please avoid lumpy citations like [1,2] or [5-7] or [4,20,21]. It is better to specify each reference.

2- The main objective of the work must be written on the more clear and more concise way at the end of introduction section.

3- I feels weird to read a paper without a proper conclusion. This is not acceptable. Please fix this.

4- The manuscript should have a comparison between this method and ther others presented in the literature. 

Comments on the Quality of English Language

Please do not use contractions like it's and don't.

Reviewer 4 Report

Comments and Suggestions for Authors

The author discussed about Effect of Multiple Reclosing Time Interval for Axial Vibration of Winding, the following points to be addressed.

1.       what is need of this study, & how important in axial vibration in power transformers

2.       Highlights the novelty of the work by points at end of introduction section.

3.       How does the reclosing of circuit breakers or switches influence the axial vibration of the transformer winding.

4.       List out key factors that contribute to axial vibration in power transformers

5.       Need more explanation about the concept of multiple reclosing time intervals and how they relate to axial vibration.

6.       How frequency and magnitude of axial vibration change during normal operation and after reclosing events.

7.       discuss and add some results related to the potential consequences of excessive axial vibration in a transformer

8.       How can finite element analysis or computational simulations be used to model and analyze the axial vibration behaviour under different reclosing scenarios.

9.       Are you using any mitigation strategies or control measures that can be implemented to reduce the adverse effects of axial vibration due to reclosing.

10.   What about any design considerations or modifications that can be made to transformer systems to minimize the impact of axial vibration during reclosing.

11.   How can transformer maintenance practices be adjusted based on the findings related to axial vibration and reclosing events.

12.   Compare your system with conventional methods with graphical representations.

13.   What are the limitations of current research or understanding in this field, and what areas require further investigation?

Comments on the Quality of English Language

Moderate corrections required 

Round 2

Reviewer 2 Report

Comments and Suggestions for Authors

The authors significantly improved the manuscript and responded to the comments included in the first review. All answers are satisfactory and the corrections introduced in the manuscript improve its scientific value.

Reviewer 3 Report

Comments and Suggestions for Authors

The authors seemed to have answered all of my comments. I don’t have further questions 

Reviewer 4 Report

Comments and Suggestions for Authors

It can be accepted for publication